# The Role of Vitamin D in Skeletal Muscle Repair and Regeneration in Animal Models and Humans: A Systematic Review

**DOI:** 10.3390/nu15204377

**Published:** 2023-10-16

**Authors:** Miguel Agoncillo, Josephine Yu, Jenny E. Gunton

**Affiliations:** 1Centre for Diabetes, Obesity and Endocrinology (CDOE), The Westmead Institute for Medical Research, The University of Sydney, Sydney 2145, Australia; 2Faculty of Medicine and Health, The University of Sydney, Sydney 2145, Australia; 3Department of Diabetes and Endocrinology, Westmead Hospital, Sydney 2145, Australia

**Keywords:** vitamin D, skeletal muscle, vitamin D receptor, muscle degeneration

## Abstract

Vitamin D deficiency, prevalent worldwide, is linked to muscle weakness, sarcopenia, and falls. Muscle regeneration is a vital process that allows for skeletal muscle tissue maintenance and repair after injury. PubMed and Web of Science were used to search for studies published prior to May 2023. We assessed eligible studies that discussed the relationship between vitamin D, muscle regeneration in this review. Overall, the literature reports strong associations between vitamin D and skeletal myocyte size, and muscle regeneration. In vitro studies in skeletal muscle cells derived from mice and humans showed vitamin D played a role in regulating myoblast growth, size, and gene expression. Animal studies, primarily in mice, demonstrate vitamin D’s positive effects on skeletal muscle function, such as improved grip strength and endurance. These studies encompass vitamin D diet research, genetically modified models, and disease-related mouse models. Relatively few studies looked at muscle function after injury, but these also support a role for vitamin D in muscle recovery. The human studies have also reported that vitamin D deficiency decreases muscle grip strength and gait speed, especially in the elderly population. Finally, human studies reported the benefits of vitamin D supplementation and achieving optimal serum vitamin D levels in muscle recovery after eccentric exercise and surgery. However, there were no benefits in rotator cuff injury studies, suggesting that repair mechanisms for muscle/ligament tears may be less reliant on vitamin D. In summary, vitamin D plays a crucial role in skeletal muscle function, structural integrity, and regeneration, potentially offering therapeutic benefits to patients with musculoskeletal diseases and in post-operative recovery.

## 1. Introduction

Vitamin D deficiency increases the risk of injury and disability and decreases quality of life in patients due to its association with sarcopenia, which is a condition that leads to the loss of muscle mass and, subsequently, the loss of muscle function and strength [1,2,3]. Hence, it is necessary to understand the role vitamin D plays in skeletal muscle function, structure, repair, and regeneration. The general criterion for vitamin D deficiency is 25(OH)vitamin D (25D) levels of less than 50 nmol/L (20 ng/mL) [4]. Levels of 25D below 20 nmol/L (8 ng/mL) are associated with altered calcium homeostasis and a substantially increased risk of osteomalacia [4]. An “optimal” level of 25D is usually considered to be above 75 nmol/L, as these are the levels of 25D that are associated with the nadir in circulating parathyroid hormone.

Vitamin D is a lipid-soluble steroid hormone required for calcium and phosphate homeostasis [5]. It is synthesized in the skin from 7-dehydrocholesterol upon ultraviolet B (UVB) exposure, converting into pre-vitamin D_3_ [5,6]. Pre-vitamin D_3_ can also be obtained from dietary sources (e.g., oily fish, some meats, and eggs). Pre-vitamin D_3_ is hydroxylated into 25D in the liver by CYP27R1 [6] and subsequently further hydroxylated into 1,25-(OH)_2_vitamin D (1,25D) by CYP27B1 [5]. 1,25D, the biologically active form of vitamin D, binds to vitamin D receptors (VDR) in target tissues to regulate the expression of genes related to calcium and phosphate homeostasis and cellular proliferation/differentiation (Figure 1) [5,6]. Circulating 1,25D is mainly formed in the kidneys, but many cell types, including macrophages and placenta, can convert 25D to 1,25D intracellularly [5]. Vitamin D signaling plays critical roles in tissues such as muscle, aiding in function, maintenance, and development [7,8].

### 1.1. Whole-Body VDR Knockout Mice

Several in vivo studies have shown that the deletion of VDR in mice exert negative effects on skeletal muscle function, structure, and development [3,9,10,11]. Mice with whole-body VDR knockout (VDRKO) have significant reductions in grip strength and endurance [3]. In addition, decreased muscle fiber size in conjunction with increased numbers of centralized nuclei were detected in myocytes, which is a possible indication of ongoing muscle regeneration in VDRKO mice [3,12]. These in vivo studies have also shown a significant downregulation in myogenic regulatory factors, calcium handling genes, sarco-endoplasmic reticulum calcium transport ATPase channels, and atrophy-related genes [2,3,9,12,13].

### 1.2. Myocyte-VDR Knockout Mice

However, the in vivo studies performed on whole-body VDRKO mice cannot determine whether vitamin D acts directly in muscle. To address this, a study was performed in mice with myocyte-specific deletion of VDR (mVDR) [2]. Significant decreases in grip strength and endurance were also present in mVDR mice, while increased muscle fiber size and the number of centralized nuclei were also seen [2]. Similar to whole-body VDRKO mice, the expression of myogenic regulatory factors, cell cycle, calcium handling genes, and sarco-endoplasmic reticulum ATPases was significantly downregulated [2]. This study confirmed that vitamin D has a direct role in the muscle as well as indirect roles via calcium and phosphate homeostasis.

Skeletal muscles have a robust ability to respond to stressors. Muscle regeneration is an important homeostatic process for maintenance and ensures the recovery of muscle function and structural integrity after injury [14,15,16]. This process is driven by the activation of skeletal muscle stem cells, also known as “satellite cells” [17,18]. Satellite cell activity is regulated by transcriptional regulators such as paired box 7 (Pax7) [19,20,21]. Activated satellite cells proliferate, which leads to the generation of committed myoblasts, vital for muscle regeneration and self-replenishment [19,20,21].

Although it has been established that vitamin D plays an important role in muscle function and maintenance, its role in muscle regeneration and repair is less well understood and is the subject of this review. This review will address this topic by exploring cell culture studies and animal and human studies that explore how vitamin D influences muscle repair and regeneration.

## 2. Materials and Methods

This systematic review was prepared in accordance with the PRISMA guidelines [22]. The key research question for this review is “What role does vitamin D have in inducing skeletal muscle repair and regeneration in animals and humans?” The review was registered with PROSPERO (CRD42023460719).

### 2.1. Search Strategies

The following databases were searched for relevant studies to include in this review:PubMed (searched in April, May, and September 2023);Web of Science (searched in September 2023);Cochrane Library (searched in September 2023);Scopus (searched in September 2023).

The following search terms were used for PubMed, Web of Science, and Cochrane libraries with the “all fields” option and in Scopus with the “article title, abstract, keywords” option:

(muscle) AND (repair OR regeneration) AND (vitamin D)

Only articles written in English were considered for inclusion.

### 2.2. Selection of Studies

One author (MA) then screened the search results. Following this, the titles and abstracts were read. The full text of potentially relevant articles for inclusion was obtained and read, and their reference lists were screened for potential additional papers for inclusion by two authors (MA and JEG) independently.

The studies to be reviewed had to meet the following criteria for inclusion: (a) Original research articles (in vitro and in vivo studies), clinical trials, randomized controlled trials, and observational studies that discussed the role of vitamin D in muscle regeneration; (b) articles that discussed the role of vitamin D in combination with other compounds in muscle regeneration. (c) Articles had to be written in English.

The review articles were not included but were checked independently for potential additional references for inclusion by the same authors (MA and JEG). The studies had to report the effects of vitamin D on muscle function, structure, repair, and regeneration. The literature search inclusion and exclusion criteria can be found in Table 1.

### 2.3. Data Extraction

The same researchers (MA and JEG) independently reviewed the full text of the papers selected and extracted the following data, depending on the type of study assessed:

For in vitro studies: (1) the cell model used; (2) the treatment or intervention administered; (3) the effects of treatment or intervention on skeletal muscle cells.

For animal studies: (1) the genetic modification/animal model used; (2) the effects of treatment or intervention on skeletal muscle function, structure, recovery, or regeneration.

For human studies: (1) number of participants; (2) participant age and/or gender; (3) treatment/surgery/intervention administered; (4) outcomes of interventions on their skeletal muscle function, structure, and recovery.

The data collected by the researchers was then further assessed and checked by the author (JY) to ensure its accuracy.

### 2.4. Quality Assessment of Included Studies

The characteristics that were assessed included the effects of vitamin D treatment/supplementation/replete diets on muscle function and morphology and how genetic deletions of specific intermediates of the vitamin D signaling pathway affect skeletal muscles.

However, not all studies included in the systematic review were able to directly address how vitamin D affects muscle repair and regeneration, but they should be able to help expand our understanding of the issue. For example, studies that talk about how vitamin D affects angiogenesis, mitochondrial respiration, or the cell cycle of myocytes, which are processes involved in skeletal muscle function and structure maintenance, were assessed. This is because those studies could help us understand how vitamin D mediates skeletal muscle repair and regeneration in animals and humans.

MA and JEG worked together to determine the suitability of articles for assessment in this systematic review. MA and JY independently assessed the studies for risk of bias using the Cochrane revised tool for assessing risk of bias (RoB 2) [23]. The scale rate studies are based on five key domains: randomization and recruitment processes, deviations from intended interventions, missing outcome data, bias in measurements of the outcome, and selection bias. A modified version of the tool was utilized where appropriate. Any disagreements in scores were resolved by discussion amongst all authors to reach a consensus.

## 3. Results

The search identified 155 papers written in English. We then identified 23 primary research articles investigating the role of vitamin D on muscle repair and regeneration in vitro and in vivo (Figure 2). Within the 23 articles selected, 8 papers explored the effects of vitamin D on muscle regeneration using animal models [24,25,26,27,28,29,30,31], 6 papers investigated this in humans [32,33,34,35,36,37], 4 papers investigated this in vitro [38,39,40,41], 2 papers used both cell and animal models [13,42], 2 papers utilized animal models and humans [43,44], and 1 paper investigated this using cell models and humans [45]. Reference lists from these papers and muscle regeneration reviews were examined to look for any additional papers, and 29 primary research articles were identified from this examination.

### 3.1. In Vitro Studies (Table 1)

#### 3.1.1. Introduction to In Vitro Experiments

Although there was initially controversy regarding the expression of VDR in muscles [46], numerous studies have confirmed the presence of VDR in both animal and human skeletal muscles with well-validated antibodies [42,47,48]. 1,25D has also been shown to regulate VDR expression and stimulate its translocation to the plasma membrane [49]. Therefore, in vitro studies that explore the molecular mechanisms of vitamin D in both animal- and human-derived skeletal muscle cells are summarised in Table 2 and will be discussed.

#### 3.1.2. Main Findings and Their Implications

Srikuea et al. (2012) confirmed the expression of *Vdr* and *Cyp27b1* in C2C12 myoblasts and myotubes using qPCR, Western blotting, and immunocytochemistry [42]. Upon treatment of C2C12 cells with 25D or 1,25D, there was a five-fold increase in *Vdr* mRNA and protein expression within 24 h. Furthermore, *Cyp27b1* siRNA knockdown in 25D-treated myoblasts led to a two-fold increase in cell numbers compared to controls. This demonstrated that *Cyp27b1* was biologically active in skeletal muscles and was important in mediating the inhibitory effects of 25D on myoblast proliferation. Overall, this study demonstrates the presence of *Vdr* and biologically active *Cyp27b1* in a cultured myoblast cell line [42].

Srikuea et al. (2020) performed another in vitro study using skeletal muscle stem cells (SMSC) isolated from male C57BL/6 mice at 4, 24, and 72 weeks of age to determine the expression of vitamin D system-related proteins in these cells after treatment with 1,25D [50]. Vitamin D system proteins were expressed in developmental, mature, and aged SMSCs. Following 1,25D stimulation, the authors demonstrated increased expression of the VDR protein in addition to co-localization of the CYP24A1 protein. However, qualitative analysis of the SMSCs showed decreased responsiveness to 1,25D with advanced age (an impaired increase in VDR after 1,25D treatment). In comparison, VDR protein was highly expressed in undifferentiated SMSCs after 1,25D treatment.

Saito et al. studied the effects of eldecalcitol (a vitamin D analogue) in C2C12 myogenic cells [51]. Eldecalcitol at 1, 10, and 100 nM yielded a dose-dependent increase in Vdr, *MyoD*, and *Igf1* gene expression. Furthermore, there was significant upregulation of myosin heavy chain (MHC) subtypes *Iia*, *Iib*, and *Iix* at 10–100 nM, while the expression of *MhcIβ* remained unchanged. Protein expression of fast MHC subtypes also increased with 1 and 10 nM eldecalcitol.

Okuno et al. investigated the effects of 1,25D on proliferating, differentiating, and differentiated C2C12 myoblasts [52]. C2C12 myoblasts were treated with 0, 1, 10, or 100 nM 1,25D. Myoblast proliferation was reduced with 1,25D treatment in a dose-dependent manner over 72 h. This was confirmed using a WST-1 proliferation assay. Furthermore, there was an increased percentage of cells in G_0_/G_1_ phases identified by flow cytometry and elevated expression of proliferation inhibitors *p21* and *p27* using qPCR. 1,25D decreased the protein expression of MHCs in a dose-dependent manner in the differentiating myoblasts. Decreased expression of *Myogenin*, MHC neonatal genes, and decreased MHC-positive myocytes were found. Finally, 1,25D enhanced the expression of fast MHCs in differentiated myoblasts. In summary, 1,25D inhibited myoblast proliferation and myogenesis in differentiated myoblasts in a dose-dependent manner, while enhancing fast myosin heavy chain protein expression in differentiated myoblasts.

Girgis et al. performed in vitro studies to address whether vitamin D had effects on C2C12 muscle cells [8,47]. To achieve this, C2C12 cells were treated with 100 nM of 25D or 1,25D for 72 h and assessed the effects of 25D and 1,25D on proliferation and differentiation [8]. They reported an increase in the expression of *Vdr*, *Cyp24a1,* and *Cyp27b1* upon treatment of C2C12 cells with either 25D or 1,25D. Functional *Cyp27b1* was identified in these cells after seeing significant, comparable luciferase reporter activity with 25D and 1,25D treatments [8,47]. Similar to the above studies, they found slower myoblast proliferation with 25D or 1,25D treatment. To assess the mechanism by which 1,25D exerts its antiproliferative effects, mRNA levels of cell cycle genes were measured. Overall, there was an increase in the expression of *Atm* and *Rb* and a reduction in *C-myc* and *Cyclin D1*. These genes are responsible for regulating the G1-S phase restriction point, which helps explain the decrease in myoblasts found in culture. There was also decreased myotube formation with both 25D and 1,25D treatments. Interestingly, despite the lower number of myotubes, individual myotubes were larger after 25D or 1,25D treatment. Consistent with that, *Myostatin* mRNA levels declined with vitamin D treatment.

Similarly, Camperi et al. investigated the role of vitamin D on skeletal muscle development using C2C12 cells [27]. Treatment with vitamin D reduced myogenic differentiation. At day 4 of differentiation, there was a marked increase in VDR protein, paralleled by a significant decrease in myogenin protein. Silencing *Vdr* with a lentivirus-delivered shRNA restored myoblast differentiation in C2C12 cells.

Hosoyama et al. performed a study to examine the effects of vitamin D on myogenesis and muscle fiber maintenance of immortalized mouse myogenic cells in culture [38]. They used Ric10 myoblasts to investigate the effects of 1 µM 1,25D on cell proliferation and differentiation. These very high concentrations of 1,25D (which normally circulate in picomolar concentrations) did not influence myoblast proliferation but did inhibit the expression of some myogenic genes, including *Myf5* and *Myogenin*. Additionally, there was inhibition of fusogenic gene expression, which correlated with reduced myotube formation, in which both myoblast-to-myoblast and myoblast-to-myotube fusion were inhibited. However, vitamin D treatment induced myotube hypertrophy (significantly increased diameter of myotubes) in individual myotubes through the activation of anabolic pathways. Thus, very high concentrations of vitamin D had both positive and negative effects on muscle cells. The authors suggest that vitamin D administration could help treat severe defects in muscle regeneration.

Using SMSCs isolated from male C57BL/6 mice, Braga et al. demonstrated that vitamin D increased cell differentiation, proliferation, and growth [13]. Incubation of the cells with 100 nM 1,25D for 7 days showed an 8-fold increase in *Vdr* mRNA and a 1.7-fold increase in VDR protein. They also found increases in the expression of myogenic markers *Myh1*, *Tnni2,* and *Tnni3*, accompanied by increased expression of pro-myogenic factors *Bmp4*, *Igf1*, and *Fgf2*. A downregulation in myostatin expression (−2.6 fold) in combination with a 1.5-fold increase in follistatin expression was also observed. Taken together, these results demonstrate that 1,25D enhanced myogenic differentiation in satellite cells, which would be predicted to stimulate muscle repair after injury.

Bass et al. also conducted an in vitro study to understand the mechanisms of myogenic regulation by transfecting lentiviral shRNA constructs in C2C12 myoblasts to knockdown VDR [12]. VDR ablation reduced myoblast proliferation and terminal differentiation, with larger numbers of VDR-knockdown myoblasts found in the G_0_-G_1_ phase of the cell cycle using flow cytometry and the fluorescent probe propidium iodide. Consistent with this, they also observed fewer myoblasts and myotubes and an increase in the number of myonuclei and DNA content.

Irazoqui et al. investigated whether VDR and p38 MAPK were involved in the signaling mechanism triggered by 1,25D in C2C12 muscle cells [53]. Lentiviral particles containing shRNA, encoded against mouse VDR, were transfected into cells to induce VDR knockdown. 1,25D treatment of C2C12 WT cells promoted a VDR-dependent increase in cells in the S-phase, followed by arrest in the G_0_/G_1_ phase. That effect was absent in the VDR knockdown cells. Thus, the presence of VDR is important to allow cell cycle arrest, which is essential for myogenic differentiation. Furthermore, 1,25D increased the activity of p21 and p27 cyclin-dependent kinase inhibitors and *Myogenin* expression in WT cells but not in the VDR knockdown, supporting that this process of leaving the cell cycle and promoting differentiation/myogenesis occurs in a VDR-dependent manner.

Garcia et al. (2013) investigated the role of vitamin D in regulating angiogenic factors in skeletal muscle cells, which are important in muscle repair and healing. They used C2C12 cells treated with 100 nM 1,25D for 1, 4, or 10 days [39,54]. Overall, they found that vitamin D upregulated the expression of *Fgf1* and V*egfa*. On the other hand, 1,25D downregulated *Fgf2* (a skeletal muscle differentiation inhibitor) and *Timp3* (an inhibitor of angiogenesis and myogenesis) expression. Overall, vitamin D upregulated the expression of *Fgf1* (a pro-angiogenic factor induced during myogenesis) and V*egfa* (a key inducer of angiogenesis). On the other hand, 1,25D downregulated negative factors, including *Fgf2* (a skeletal muscle differentiation inhibitor) and *Timp3* (an inhibitor of angiogenesis and myogenesis) expression. Therefore, vitamin D may play an important role in promoting angiogenesis in skeletal muscle.

Garcia et al. (2011) also examined the effects of 1,25D on myoblast proliferation, progression, and differentiation into myotubes [54]. C2C12 myoblasts were incubated with or without 1,25D for 4 and 7 days. Proliferating cell nuclear antigen (PCNA) expression by western blotting and immunocytochemistry was used to evaluate cell proliferation. There were no significant differences in PCNA expression at 4 days, but at 7 days, PCNA expression was reduced in 1,25D-treated myoblasts. After this, the effects of 1,25D on myogenic differentiation were assessed. There was increased expression of myogenic markers such as MyoD+ and desmin and an increased diameter of MHC Type II fibers with the 1,25D treatment, indicating that it enhances myogenic differentiation. Furthermore, several promyogenic factors, such as *MyoD*, *Desmin*, *Myogenin*, and *Igf-II*, were upregulated in 1,25D-treated myoblasts, while there was a downregulation in the expression of *Mstn,* which encodes myostatin. Finally, the effects of 1,25D on Fst (a protein that inhibits myostatin activity) were assessed. 1,25D treatment significantly increased *Fst* mRNA and protein levels in myoblasts. Together with reduced *Mstn,* this should facilitate muscle growth.

Mizutani et al. explored the effects of vitamin D on C2C12 myoblasts and myotubes with 100 nM and 1 µM 1,25D treatments for 24 h [40]. They performed a microarray analysis on the C2C12 myoblasts and myotubes to study the changes in gene expression with vitamin D treatment. Treatment with vitamin D upregulates genes responsible for promoting angiogenesis (*Vegfa*), myocyte differentiation (*Mdfi*), muscle hypertrophy (*Igfbp3*, *Igfbp5*, and *IGF1*), and lipid metabolism (*Dgat1*, *Dgat2*, *lipin3*, and *Lpl*). Together, these findings suggest vitamin D plays an important role in processes mediating skeletal muscle function and repair.

Salles et al. investigated the combined effects of 1,25D with insulin and leucine on the protein synthesis rate in C2C12 cells [55]. Following 5 days of differentiation, myotubes were treated with 0, 1, or 10 nM 1,25D for 72 h, followed by stimulation with insulin (100 mM) plus leucine (5 mM) for 30 min. Control cells were left serum-free and leucine-free. Interestingly, 1,25D enhanced protein synthesis, with significant increases only noted with 10 nM, independent of insulin and leucine treatment. 1,25D exposure also enhanced Akt/mTOR pathways and significantly increased the gene and protein expression of VDR and insulin receptors.

Buitrago et al. investigated the modulation of Akt by 1,25D at the proliferation and early differentiation stages of C2C12 cells [56]. Akt phosphorylation was increased during both proliferation and early differentiation through Src and PI3K activation with 1,25D treatment. 1,25D stimulated a 40% increase in the proliferation of C2C12 myoblast cells through PI3K activation. It also stimulated the phosphorylation of Akt and the expression of myosin heavy chains and myogenin during early-stage differentiation of C2C12 cells.

Van de Meijden et al. investigated the effects of 25D and 1,25D on myoblast proliferation, differentiation, and size and whether vitamin D metabolism occurred in C2C12 myoblasts [57]. C2C12s were cultured and treated with 1000 nmol/L 25D, 100 nmol/L 1,25D, or 0.1% ethanol (vehicle) for 24, 72, or 96 h. Increased *Cyp24* and *Vdr* levels were found, while decreased *Cyp27b1* and *Myogenin* levels were found using RT-qPCR in proliferating myoblasts after 72 and 96 h. Furthermore, increased *Myhc-1 -Iia*, *Iib*, *and Iix* mRNAs were found. Increased myotube diameter was detected in 25D-treated myoblasts after 96 h. To understand the effects of 25D on myotube size, the effects of 25D and 1,25D on the Akt/mTOR signaling pathway components were investigated using western blotting. No changes in p-Akt, p-6, or total P6 levels were detected, but significant increases in total Akt levels were found. Finally, myotubes did not synthesize detectable 1,25D after exposure to 25D, but increases in *Cyp24a1* mRNA were detected in myoblasts and myotubes upon 25D treatment. These results show that C2C12 cells respond to both 1,25D and 25D.

Ashcroft et al. showed that 1,25D directly regulated mitochondrial function using lentiviral-induced VDR-KD C2C12 myoblasts [58]. In contrast, 1,25D decreased the expression of proteins associated with mitochondrial fission (Fis1 and DRP1), phosphorylated pyruvate dehydrogenase, and PDK4. Finally, 1,25D did not affect the expression of proteins in the mitochondrial respiratory chain.

Schnell et al. performed a study to investigate the mechanism through which vitamin D could regulate mitochondrial function in C2C12 cells with 100 nM calcitriol [41]. Vitamin D treatment was shown to enhance VDR and PLIN2 expression and genes related to lipolysis and β-oxidation. It also increased mitochondrial metabolism in C2C12 myotubes. Following this, siRNA-induced PLIN2 knockdown was performed to investigate its importance in vitamin D-mediated intramyocellular lipid accumulation and mitochondrial function. PLIN2 was selected for investigation as it is most highly expressed in skeletal muscles and has been shown to increase oxidative capacity and metabolic function [59,60]. PLIN2 knockdown had no effect on lipid accumulation in C2C12 myotubes, but it decreased the expression of genes related to lipolysis and lipid storage. As a result, it reduced the mitochondrial oxygen consumption rate. Hence, this study highlighted that the ability of vitamin D to enhance mitochondrial activity in skeletal muscles was partially dependent on PLIN2.

**Table 2 nutrients-15-04377-t002:** In vitro studies of vitamin D treatment on murine or human derived skeletal muscle cells.

Reference	Cell Model	Treatments	Notes and Effects
[42] Srikuea (2012)	C2C12	25D (2 µM) or 1,25D (20 nM) + CYP27B1 siRNA	↑ *Vdr* mRNA with 25D and 1,25D treatment. Cyp27B1 knockdown caused and ↑ cell numbers with 25D treatment, suggesting inhibition by 25D
[50] Srikuea (2020)	C57BL/6 mouse SCs (male)	100 nM 1,25D	↑ VDR and CYP24A1 proteins in developmental, mature, and aged cells. ↓ responsiveness of SMSCs and ↓ VDR expression with aging
[51] Saito	C2C12	Eldecalcitol 1, 10 or 100 nM	↑ *Vdr*, *MyoD* and *Igf1* mRNAs, ↑ expression of *MHC Iia*, *Iib* and *Iix* at 10–100 nM, ↑ fast myosin head chain proteins at 1 and 10 nM
[52] Okuno	C2C12	1,25D at 1, 10 or 100 nM	↓ myoblast proliferation in differentiating phase ↑ fast myosin head chains in differentiated phase
[8] Girgis	C2C12	100 nM 25D or 100 nM 1,25D	25D and 1,25D caused ↑ *Vdr*, *Cyp27b1* and *Cyp24a1* mRNAs, ↑ genes involved in G0/G1 arrest, ↓and in G1/S transition genes. ↓ myotube formation and myogenic regulatory factors, ↓ myostatin and ↑ myotube cross-sectional size with 1,25D
[27] Camperi	C2C12 myoblasts	1,25D (10 or 100 nM)	↓ myoblast proliferation and myogenic differentiation. ↑ VDR and myogenin protein. Lentiviral αVDR shRNA restored myoblast differentiation
[38] Hosoyama	Immortalis-ed mouse cells	1 µM 1,25D	↓ expression of myogenic regulatory factors, *Myf5* and myogenin in proliferating myoblasts. ↓ myoblast-to-myoblast and myoblast-to-myotube fusion ↑ hypertrophy and protein anabolism
[13] Braga	C57BL/6 mouse SCs	100 nM 1,25D	↑ myogenic effect on satellite cells, ↑ myotube formation ↓ expression of MSTN
[12] Bass	C2C12	Lentiviral anti-VDR shRNA	↓ total protein content, lower myofiber area, ↓ myogenesis ↓ mitochondrial respiration related proteins and genes Activation of autophagic processes, no effect on muscle protein synthesis or anabolic signalling
[53] Irazoqui	C2C12	Lentiviral anti-VDR shRNA	↑ VDR and p38-dependent S-phase peaks, ↑ p21 and p27 cyclin-dependent kinase, ↑ myogenin expression. Induced cell arrest at G0/G1 phase and ↑ myogenic differentiation.
[39] Garcia (2013)	C2C12	100 nM 1,25D	↑ *Fgf1* and *Vegfa* expression, ↓ *Fgf2* and *Timp3* expression Vitamin D regulated angiogenic factors in skeletal muscle cells
[54] Garcia (2011)	C2C12	100 nM 1,25D	↓ PCNA expression, ↑ *MyoD*, *Desmin*, *Myogenin* and *Igf-II*, *↓ Mstn*, *↑Fst* Vitamin D enhanced myoblast differentiation
[40] Mizutani	C2C12 myoblasts and myotubes	100 nM and 1 µM 1,25D	↑ expression of genes involved in promoting angiogenesis, myoblast differentiation, muscle hypertrophy and lipid metabolism
[55] Salles	C2C12	100 mM insulin + 5 mM leucine and 1,25D (0, 1 or 10 nM)	1,25D enhanced the protein anabolic effects of insulin and leucine in myotubes in a dose-dependent manner
[56] Buitrago	C2C12 myoblasts	1 nM 1,25D	↑ Akt phosphorylation during both proliferation and early differentiation, ↑ expression of myosin head chains and myogenin, ↑ myoblast proliferation
[58] Ashcroft	C2C12 myoblasts	VDR-KD lentiviral plasmids	↓ mitochondrial respiration, ↓ mitochondrial ATP production No changes in mitochondrial protein content, and markers for mitochondrial fission
[57] van der Meijden	C2C12 myotubes and myoblasts	1 mM 25D or 100 nM 1,25D	*↑ Cyp24* and *Vdr* levels, ↓ *Cyp27b1* and *Myogenin*, *↑ Myhc-1 -Iia*, *Iib* and *Iix*, ↑ Increased myotube diameter with 25D. No changes in p-Akt, p-6 and total P6 levels, ↑ Total Akt with 25D and 1,25D, *↑ Cyp24a1* mRNA with 25D in myotubes and myoblasts.
[61] Olsson	Human myoblasts	25D (100 nM) or 1,25D (1 nM or 100 nM)	↑ *VDR* and *CYP24A1* mRNA and proteins, ↓ cell proliferation, ↓ myotube formation, ↓ expression of myogenic regulatory factors ↓ cell cycle regulators, ↑ Cdk inhibitors *p27* and *p21*, *FOXO3*, *↑* Notch signalling pathway components with 100 nM 1,25D.
[62] Ryan	Human skeletal muscle cells	10 nM 1,25D	↑ Increased oxygen consumption, ↑ mitochondrial volume and branching, ↑ expression of OPA1 and Mfn1, ↓ expression of mitochondrial fission proteins, ↓ phosphorylated pyruvate dehydrogenase and PDK4
[41] Schnell	C2C12 myotubes	100 nM Calcitriol ± PLIN2 siRNA	↑ VDR and PLIN2 expression with calcitriol treatment, ↑ SDH activity with calcitriol treatment, ↑ intramyocellular lipid accumulation with calcitriol treatment, ↓ lipolysis and mitochondrial respiration with vitamin D + PLIN2 siRNA treatment
[63] Romeu Montenegro	Human skeletal muscle cells	100 nM 1,25D	↓ myocyte proliferation, ↑ myoblast differentiation, ↑ Akt and mTOR signalling cascade activity, ↑ mitochondrial oxygen consumption
[64] Hayakawa	Human myotubes	10 nM 1,25D	↓ expression of *MAFbx* and *MuRF1*, ↑ expression of IL-6 ↓ protein phosphatase 2A (PP2A) in human myotubes, ↑ AKT-1
[45] Owens	Human myoblasts	1, 25D (10 or 100 nM)	Improvements in myotube fusion and differentiation with 10 and 100 nM 1,25D. ↑ expression of *MRF4*, *MYOG* and *VDR*, ↑ myotube diameter, area, and number of myotubes per field.

SC—satellite cells.

Outcomes for human skeletal muscle cells are also shown in Table 1. Olsson et al. [61] investigated the effects of vitamin D in human muscle cells at various stages of differentiation. They isolated myoblasts for cell culture and snap-froze muscle tissues biopsied from resting human skeletal muscles for their experiments. They found increased *VDR*, *CYP27B1*, and *CYP24A1* mRNA and protein expression in myoblasts and myotubes compared to adult skeletal muscles. Treating human myoblasts with 100 nM 1,25D resulted in significant increases in *VDR* and *CYP24A1* mRNA and VDR protein, compared to the control group and those treated with 100 nM 25D. Cell proliferation was also inhibited by 1,25D without affecting cellular apoptosis or necrosis. Human myotube formation was inhibited by 1,25D treatment without altering endogenous myostatin expression. Global gene expression in biopsied muscle was performed using microarray analysis. Decreased expression of myogenic regulator factors (*MYOD*, *MYOGENIN*, and *MEF2C*), cell cycle regulators (*Cyclin A2*, *B1*, *D3*, and *E1*), and muscle structural proteins (*TNNI1*, *TNNT1*, *TNNC1*, *TNNC2*, *MHCI*, and *MHCII*) was seen with 1,25D. In contrast, cyclin-dependent kinase inhibitors *p27* and *p21*, *FOXO3*, and components of the Notch signaling pathway were increased, further emphasizing the role of 1,25D on myoblast differentiation and proliferation. Gene ontology analysis found that pathways associated with myoblast differentiation and cell fate commitment were regulated by 1,25D.

Ryan et al. examined the mechanism of action of 1,25D in skeletal muscle using human skeletal muscle cells [62]. The effects of 1,25D treatment on mitochondrial oxygen consumption, dynamics, biogenesis, and changes in the expression of nuclear genes encoding mitochondrial proteins were assessed. Increased oxygen consumption was detected in adult skeletal muscle cells with 1,25D treatment, which was dependent on the presence of VDR. In addition, 1,25D increased mitochondrial volume, branching, and OPA1 protein expression.

Romeu Montenegro et al. investigated the effects of 100 nM vitamin D treatment on myocyte proliferation, differentiation, protein synthesis, and biogenesis at 24, 48, 72 h, and 5 days [63]. Decreased myocyte proliferation was found at 24, 48, and 72 h after treatment, with a 30% decrease in the number of cells found, a decreased number of cells in the G2/M phase, and decreased BrdU incorporation. On the other hand, vitamin D treatment significantly enhanced myoblast differentiation, accompanied by an increase in expression of differentiation markers (myogenin, troponin T type 1) and a decrease in myosin heavy chain 2 levels. The reported effects of vitamin D on myocyte proliferation and differentiation are consistent with the other in vitro studies reviewed [52,54,56,61]. In addition, enhanced protein synthesis was observed with an increase in Akt and mTOR downstream signaling cascade activity after vitamin D treatment. Increased mitochondrial oxygen consumption was also observed, along with enhanced maximal respiration and ATP production observed in myoblasts and myotubes. Overall, this study explored how vitamin D affected the different biological processes in skeletal muscles to maintain their structural integrity and quality, which provided us with insights on the possible mechanisms by which it mediates muscle repair and regeneration.

Hayakawa et al. studied the effects of 1,25D on changes in the expression of genes related to skeletal muscle atrophy and hypertrophy [64]. Decreased expression of *MAFbx* and *MuRF1*, muscle-specific ubiquitin ligases, was found after 1,25D treatment for 24 and 72 h. Hence, the authors suggest that 1,25D suppresses skeletal muscle degradation. On the other hand, increased expression of interleukin 6 (*IL6*) was observed. Interleukin-6 promotes muscle growth and myogenesis by acting on satellite cells [65]. 1,25D also inhibited protein phosphatase 2A (PP2A) in human myotubes, which helped to upregulate AKT-1. This process also helped to induce muscle hypertrophy and downregulate the expression of *MuRF1* and *MAFbx*. Hence, their study demonstrated that 1,25D suppressed genes involved in promoting muscle atrophy while enhancing the expression of muscle growth and myogenesis promoters.

Owens et al. recruited 14 male volunteers with low serum 25D (37 ± 11 nmol/L) and performed biopsies on them to isolate myoblast cells [45]. After biopsy, they subjected the myoblasts to mechanical injury for in vitro studies on muscle repair, regeneration, and hypertrophy in the presence or absence of 10 or 100 nmol 1, 25D. They found improvements in myotube fusion and differentiation at the biochemical, morphological, and molecular levels after treatment with 10 nM 1,25D. There were significant increases in gene expression of *MRF4*, *MYOG*, and *VDR* [45]. Furthermore, they found increases in myotube diameter, area, and number of myotubes per field.

#### 3.1.3. Conclusions: In Vitro Studies

Overall, the cultured cell studies show vitamin D plays important roles in myoblast growth and, as expected for a transcription factor, regulates gene expression. They also showed that enhancing vitamin D signaling inhibits cell proliferation but appears to produce larger, possibly better-differentiated, muscle cells. This review also explored studies that investigated the effects of vitamin D on angiogenesis, mitochondrial activity, its signaling pathway components, and the cell cycle. These processes play an important role in maintaining skeletal muscle function and morphology and collectively show the critical role vitamin D plays in skeletal muscle repair and regeneration.

However, the studies included in this review mainly focus on exploring the effects of vitamin D on myoblasts and myotubes, while limited studies focus on its effects on satellite cells. It is important to better understand how vitamin D affects satellite cells as they promote myocyte proliferation and differentiation, key processes for initiating skeletal muscle repair and regeneration. In this review, only two articles directly addressed this topic in cell culture. Hence, more cell studies addressing the effects of vitamin D on satellite cells could be performed.

In addition, the nature of tissue culture experiments means that they cannot directly address questions of muscle regeneration. The next section will therefore discuss animal studies.

### 3.2. Animal Studies (Table 3)

#### 3.2.1. Introduction to Animal Models

While the in vitro studies investigated the molecular mechanisms of vitamin D in skeletal muscle cells, the animal studies addressed its direct effects on skeletal muscle repair and regeneration. Animal studies investigating the role of vitamin D in muscle injury and regeneration are summarized in Table 3.

#### 3.2.2. Muscle Injury

In the same report discussed above for their C2C12 studies, Bass et al. investigated the potential role for VDR in muscle atrophy by performing an electro-transfer of anti-VDR lentiviral-shRNA into the tibialis anterior of rats [12]. Seven days after electrotransfer, rats were given D_2_O to enable measurement of their muscle protein synthesis. VDR knockdown induced skeletal muscle fiber atrophy, particularly in type IIB fibers. The tibialis anterior weights were not reported. Interestingly, the decrease in fiber size was not related to impaired protein synthesis; rather, it was associated with upregulated autophagic processes. This was confirmed by RNA-seq in addition to a downregulation of mitochondrial metabolism genes.

Nakamura et al. aimed to understand the mechanism of vitamin D in protecting against immobilization-induced muscle atrophy in mice [24]. Low vitamin D diet feeding in WT mice decreased muscle fiber cross-sectional area and volume and upregulated *Atrogin-1* and *MuRF1* genes in the control limb. Similar changes in these parameters were also noted in immobilized limbs in low-vitamin D-fed control mice. Refeeding these mice with a standard vitamin D diet or administering a vitamin D analogue (ED71) helped to reverse these effects in the stapled hind limb. To further establish the role of vitamin D in muscle atrophy, the authors utilized 3 mouse models: VDR knockout (VDRKO), mVDR (using the MCK-Cre), and neural-crest-specific VDR knockouts (using myelin protein zero Cre mice) and respective controls. Significant decreases in grip strength, muscle cross-sectional area, muscle volume, and upregulation of *Atrogin-1*, *MuRF1*, *Tnfa*, *Il-6*, and *Il-1b* were found in global VDRKO mice.

Nakamura and colleagues compared muscle composition and expression of atrophy-related genes in myocyte-VDR (mVDR) and neural crest-VDR knockout mice, following the same immobilization and denervation protocol [24]. Interestingly, their neural crest-VDR mice had increased muscle weights but more pronounced reductions in muscle mass following immobilization. Both mVDR and their floxed controls also experienced muscle atrophy after immobilization, which was numerically, but not significantly, greater in the mVDRs. The authors concluded that vitamin D protects against immobilization-induced muscle atrophy by acting on neural-crest-derived cells. Fiber cross-sectional areas were significantly decreased in neural crest-VDRs but not significantly in mVDRs. Thus, the authors suggest that vitamin D signaling in neural-crest-derived cells is important for diminishing muscle atrophy. However, despite the differences in statistical significance between neural crest VDRs and mVDRs, we note that the absolute decreases in muscle weights induced by immobilization appear nearly identical. Similarly, the absolute differences for fiber cross-sectional area appear similar. Additionally, for fiber cross-sectional area, we note these are not normally distributed and should be presented as medians rather than means and should be analyzed by non-parametric testing.

Panda et al. developed a mouse model of vitamin D-responsive rickets in which mice are small and have markedly reduced muscle mass by causing 1α-hydroxylase deficiency (CYP27B1 KO) [66]. It is important to note that the authors did not report on muscle function or size. However, other affected parameters that highlight the importance of vitamin D include decreased body weight, serum Ca^2+^ and PO_4_^3−^ levels, increased serum 25D, parathyroid hormone, serum alkaline phosphate, and urinary PO_4_^3−^ levels. Histological assessment found typical features of rickets such as inadequate cartilage mineralization, reduced osteocalcin, and osteoclast size and number. Yu et al. subsequently investigated the same mouse model [28]. They found decreased grip strength and muscle fiber size and decreased expression of genes related to mitochondrial metabolic activity and antioxidant enzymes. Following this, they caused a muscle injury by injecting BaCl_2_ into the tibialis anterior. Muscle weight at 7 days was not reported, but there were decreased proliferating cells (BrdU+) and centrally nucleated myofibers, indicating reduced regenerating fibers, accompanied by reduced expression of myogenic factors *MyoD*, *MyHC*, and *Myf5*. These findings suggest that 1,25D plays a role in preventing age-related sarcopenia, and adequate muscle 1,25D levels are important for skeletal muscle regeneration [66]. The authors suggested that 1,25D deficiency promoted muscle senescence.

Girgis et al. characterized the muscle phenotype and gene expression of mice with vitamin D receptor deletion (VDRKO) compared to their wild-type littermates and compared mice with a normal diet to mice fed a vitamin D-deficient diet [3]. VDRKO and vitamin D-deficient mice had significantly weaker grip strength compared to their control littermates, which progressed as the mice aged (VDRKO) or the duration of diet grew longer. VDRKO mice had significantly smaller muscle fibers that displayed hypernuclearity. Finally, VDRKO mice had decreased expression of myogenic regulatory factors, calcium handling genes, and sarco-endoplasmic reticulum transport ATPase (SERCA) channels, while they had increased expression of atrophy-related genes *MuRF1* and *Myostatin*. These findings show that vitamin D plays a role in muscle function and morphology, but because the models are whole-body, they could not determine whether the effects were direct within muscle or indirect via an effect on other tissues.

To address that limitation, Girgis and colleagues investigated the effects of deleting the vitamin D receptor in myocytes on the muscle phenotype and gene expression in mice [2]. Myocyte-specific vitamin D receptor knockout mice (mVDR) were generated by crossing human skeletal actin (HSA)-Cre mice with floxed VDR mice. The mVDR animals had reduced voluntary running speed, distance, and grip strength, while they had an increased proportional fat mass. mVDR muscle fibers showed a larger cross-sectional diameter along with the presence of increased numbers of angular fibers and centralized nuclei, which indicate ongoing remodeling. However, no clear differences in fiber types or fibrosis were observed. Finally, there was decreased expression of cell-cycle genes, cyclin-dependent kinases, calcium handling genes, and *Calbindin* mRNA and myostatin levels in mVDR. This study demonstrates the importance of direct vitamin D signaling in muscle function. However, regeneration after injury was not examined.

#### 3.2.3. Disease-Related Animal Models

Cheung et al. (2020) investigated whether vitamin D repletion would improve adipose tissue and muscle metabolism in a mouse model with chronic kidney disease (CKD)-related cachexia [26]. Six-week-old male C57BL6 mice were used and surgically induced CKD by 5/6 nephrectomy. Controls underwent a sham operation. Following surgery, they were treated with vehicle or with both 75 µg/kg/day 25D and 60 ng/kg/day 1,25D via a subcutaneous osmotic pump for 6 weeks. As 1 µg of 25D is equivalent to 40 IU, this dose of 25D is very high; it corresponds to 210,000 IU per day in a 70 kg person, which is 525 times the recommended daily intake of vitamin D. The 1,25D dose is also high; 60 ng/kg would be 4200 ng for a 70 kg person, with a standard dose being 250–500 ng daily in people. It is therefore unexpected that they reported no significant increase in 25D or 1,25D levels in sham mice. However, treatment did prevent the decreased levels seen with nephrectomy.

Treatment with 25D and 1,25D normalized ATP content in the muscles of CKD mice. Furthermore, vitamin D repletion improved muscle fiber size, improved muscle function (grip strength and rotarod activity), decreased expression of profibrotic genes, and increased anti-fibrotic gene expression in CKD mice. Lastly, vitamin D repletion helped to alleviate muscle fat infiltration and improve skeletal muscle mass regulation signaling in CKD mice. Overall, their findings suggested that vitamin D repletion helped to decrease muscle wasting in CKD mice.

Cheung et al. (2020a) also examined whether vitamin D repletion ameliorated adipose tissue browning and muscle wasting in mice with deletion of cystinosin (Ctns^−/−^) [25]. To achieve this, they treated 12-month-old Ctns^−/−^ and WT control mice (obtained from separately sourced C57Bl/6 mice) with vehicle or 25D + 1,25D (75 µg/kg/day and 60 ng/kg/day, respectively) for six weeks. Normalized muscle function (grip strength and rotarod test), muscle fiber size, collagen content and energy expenditure were noted in Ctns^−/−^ mice after vitamin D supplementation. Supplementation also normalized muscle ATP content in Ctns^−/−^ mice, with levels comparable to the WT group. Gene expressions of *Igf1*, *Pax7*, *MyoD*, *Murf1*, and *Myostatin* in Ctns^−/−^ mice that received vitamin D were also comparable to WT controls. Finally, differential expression of genes implicated in cellular processes, biological regulation and metabolic processes were detected using muscle RNA-seq analysis.

Camperi et al. investigated the effects of vitamin D treatment on cancer-induced muscle wasting in rats [27]. They found a decrease in circulating vitamin D levels in tumor-bearing rats compared to controls, with a corresponding upregulation of *Vdr* mRNA in their muscles. After daily vitamin D administration (80 IU/kg of cholecalciferol by gavage) until sacrifice at 7, 14, or 28 days, tumor-bearing rats had increased muscle *Vdr* expression without significant change in the cancer-induced reductions in body weight, gastrocnemius, or tibialis anterior size. Similar results were seen with two other different tumor models (colon carcinoma and lung carcinoma). Muscle function was not reported.

**Table 3 nutrients-15-04377-t003:** Vitamin D studies in animal models.

Reference	Model/Genetic Modification	Notes and Effects
[12] Bass	VDR knockout via lentiviral shRNA electro-transfer	↑ skeletal muscle atrophy, ↑ autophagic processes ↓ mitochondrial metabolism. No changes to anabolic signalling or muscle protein synthesis
[24] Nakamura	VDR knockout	↓ grip strength, ↓ muscle cross-sectional area, ↓muscle volume, *↑ Atrogin-1*, *↑ MuRF1*, *↑ Tnfa*, ↑ *Il-6* and ↑ *Il-1b*
	mVDR knockout	↓ muscle fiber cross-sectional area. Muscle atrophy upon immobilisation.
	neural crest- specific VDR knockout	↑ muscle weight compared to mVDR mice, ↓ muscle weight upon immobilisation, ↓ muscle fiber cross-sectional area
[28] Yu	CYP27B1 knockout	↓ grip strength, ↓ muscle fiber size, ↓ *MyoD*, *MyHC*, *Myf5* expression, ↓ BrdU positive cells
[66] Panda	CYP27B1 knockout	↓ Body weight, ↓ serum Ca^2+^ and PO_4_ ↑ serum 25D, PTH, ↑ serum alkaline phosphate, ↑ urinary PO_4_^2− s^
[3] Girgis (2015)	VDR knockout and vitamin D deficient mice	↓ grip strength, ↓ muscle fiber size, ↓ myogenic regulatory factors, ↑ myostatin levels, ↓ calcium handling genes, ↓ sarco-endoplasmic reticulum calcium transport ATPase channels, ↑ atrophy-related gene *MuRF1*
[2] Girgis (2019)	myocyte-specific vitamin D receptor knockout mice	↓ voluntary running speed, distance, and grip strength, ↑ muscle fiber diameter, ↓ cell-cycle genes, ↓ cyclin-dependent kinases, ↓ calcium handling genes, calbindin, ↓ myostatin. No differences in muscle metabolism and fiber type observed. No increase in fibrosis.
[25] Cheung (2020a)	Cystinosis knockout	Normalized muscle function, muscle fiber size, collagen content, energy expenditure, muscle ATP content with vitamin D supplementation. Normalized expression of *Igf1*, *Pax7*, *MyoD*, *Murf1* and *myostatin*.
[26] Cheung (2020b)	5/6 nephrectomy, rats	Normalized muscle fiber size, ↑ muscle function ↓ expression of profibrotic genes, ↑ expression of anti-fibrotic genes, ↓ muscle fat infiltration, ↑ skeletal muscle mass regulation. Normalized muscle ATP content
[27] Camperi	Tumor-transplant, rats	↓ circulating vitamin D levels, ↑ *Vdr* mRNA in tumor-bearing rats. ↑ *Vdr* mRNA expression after vitamin D administration. No significant change in body weight, gastrocnemius, or tibialis anterior size.
[42] Srikuea	BaCl_2_ induced injury, mice	↑ *Vdr* and *Cyp27b1* expression following injury
[31] Stratos	Crush injury, rats	↑ cell proliferation, ↓ apoptosis with high dose vitamin D. No changes in twitch strength, ↑ tetanic strength, no changes in satellite cell number and myocyte size with high dose vitamin D.
[29] Srikuea and Hirunsai	BaCl_2_ induced injury, mice	↑ *Vdr* expression, regulated *Cyp24a1* and *Cyp27b1* levels. ↓ cross-sectional area of regenerating skeletal myocytes, ↓ satellite cell differentiation, ↓ regenerative fiber formation No effect on regenerating muscle weight and fiber typing
[30] Dominguez-Faria	Old Wistar Rats	↓ plasma 25D and tibialis anterior weights in vitamin D deplete diet fed rats, ↓ cell proliferation and notch signalling pathway genes, ↓ *Bmp4*, *↓ Fgf2*, *↓ PCNA* protein expression, ↓ *Hes1*
[67] Bhat	Male rats Vitamin D deficient diet	↓ muscle weight, ↓ lean body mass, ↓ Type II muscle fiber area, ↑ muscle protein degradation, ↓ muscle protein synthesis, ↑ *Atrogin-1* and *MuRF1*, *↑ Psc2 and Psc8*, *↓ MyoD*, *MyoG and Myf5*
[44] Bang	Rats	↓ VDR expression in VDD group muscles at both 16 and 32-week timepoints, ↓ muscle fiber cross-sectional areas and volumes.
[43] Bass	In vivo electro-transfer of lentiviral VDR	↑ muscle protein synthesis, ↑ expression anabolic signalling intermediates, ↑ hypertrophy, ↑ extracellular remodelling, ↑ satellite cell content, ↑ myofiber cross-sectional area

#### 3.2.4. Injury Studies

Using mice, Srikuea et al. (2012) found an increase in *Vdr* and *Cyp27b1* expression in regenerating skeletal muscles following tibialis anterior injury induced by BaCl_2_. Mice were sacrificed 1 week after injury [42]. Muscle weights and functions were not reported.

Stratos et al. investigated whether vitamin D improved muscle healing after muscle injury by inducing crush injury of the soleus muscles in male rats [31]. After crush injury, the rats received vehicle, or “low dose”, vitamin D at 0.83 mg/kg or a high dose at 8.3 mg/kg. Even the low dose is an extreme dose; it corresponds to 33,000 IU/kg, or 2,324,000 IU for a 70 kg person. As mentioned above, the recommended daily supplement for people is 400 IU. In humans, doses of ‘only’ 250,000–300,000 IU are associated with short-term *increased* falls and fracture risks. Levels of 25D in the rats were consistently 4–6 times the controls.

High-dose vitamin D increased cell proliferation and reduced apoptosis 4 days after injury. No changes in satellite cell numbers were found. High-dose D caused no changes in muscle twitch strength at any time but was associated with a ~15% increase in muscle tetanic strength seen only at 42 days. No changes in myocyte size were reported. Srikuea and Hirunsai assessed the effects of intramuscular administration of 1,25D on regenerative capacity, muscle fibrosis, and angiogenesis in mice [29]. They injected 1.2% BaCl_2_ into the tibialis anterior to induce muscle damage. After 4 to 7 days, mice were given either 1 µg/kg estimated tibialis anterior weight of 1,25D or 1 µg/kg relative to mouse body weight of 1,25D or vehicle [29]. 1,25D treatment increased VDR expression and regulated Cyp24a1 and Cyp27b1 levels. It decreased the cross-sectional area of regenerating skeletal myocytes, decreased satellite cell differentiation, and decreased regenerative muscle fiber formation. There was no effect on regenerated muscle weight and no change in fiber typing. Muscle function was not assessed.

#### 3.2.5. Vitamin D Deficiency

Domingues-Faria et al. used 15-month-old male rats fed a vitamin D-deficient diet for 9 months to investigate the effects of vitamin D deficiency on skeletal muscle [30]. They reported significant decreases in plasma 25D concentrations in depleted rats and significant drops in tibialis anterior weights. Following this, they performed qPCR and Western blotting on their tibialis anterior and found decreases in the levels of cell proliferation and Notch signaling pathway genes, including *Bmp4* (−39%), *Fgf*-2 (−31%), a reduction in PCNA protein expression (−56%), and the downregulation of cleaved Notch protein and its target *Hes1* (−35%). Muscle function was not reported. They concluded that vitamin D deficiency promotes skeletal muscle atrophy.

Bhat et al. examined the involvement of the ubiquitin-proteasome and other proteolytic pathways in vitamin D deficiency-induced muscle atrophy using male rats [67]. Rats were fed with either a vitamin D-deficient or control diet for 12 weeks, after which the vitamin D-deficient rats were switched to either a control diet or a high-calcium diet or continued their vitamin D-deficient diet for another 6 weeks. Vitamin D deficiency led to significant decreases in muscle weight, lean body mass, and type II muscle fiber area, which could only be corrected by vitamin D supplementation. Furthermore, it accelerated muscle protein degradation and decelerated muscle protein synthesis, which was partially corrected by high-calcium diets. Accelerated protein breakdown and decelerated protein synthesis induced by vitamin D deficiency were found to be associated with increased expression of atrophy-related genes (*Atrogin-1* and *MuRF1*) and proteasomal subunit genes (*Psc2* and *Psc8*) and decreased expression of myogenic genes (*MyoD*, *MyoG*, and *Myf5*), respectively. Furthermore, increased expression of the E2-ubiquitin conjugating enzyme and ubiquitin conjugates was detected. Increased chymotrypsin-like, trypsin-like, and caspase-like activities were found in muscle extracts. Finally, no changes in lysosomal and calpain enzyme activities were detected. Muscle function was not reported.

Bang et al. investigated the relationship between paraspinal muscle changes and serum vitamin D levels in rat models [44]. Seventy-five adult male rats were randomly and blindly allocated into three groups: control, vitamin D deficient (VDD), and vitamin D-deficient replacement (VDDR). Vitamin D deficient diets were given to the VDD group for 32 weeks, while the VDDR group rats were given a normal control diet 16 weeks after being on vitamin D-deficient diets. Serum 25D levels were significantly lower in the VDD and VDDR groups at the 8- and 16-week timepoints compared to the controls. Vitamin D supplementation in VDDR rats showed serum 25D levels that were comparable to controls by 24 weeks, while the VDD group 25D levels remained low until the 32-week timepoint. VDR expression was significantly lower in VDD group muscles, and their muscle fiber cross-sectional areas and volumes were significantly smaller at both 16 and 32-week timepoints.

#### 3.2.6. VDR Overexpression

To understand the role of VDR upregulation on muscle mass and health, Bass et al. performed an in vivo electro-transfer of lentiviral VDR into the tibialis anterior of rats and followed the animals for 10 days [43]. VDR overexpression (VDR-OE) was confirmed by qRT-PCR, western blotting, and immunohistochemistry. They found VDR-OE increased muscle protein synthesis. An increase in the expression of anabolic signaling intermediates was also observed. Additionally, RNA-seq data confirmed VDR overexpression upregulated hypertrophy and extracellular remodeling. Finally, they saw an increase in satellite cell content and myofiber cross-sectional area using immunofluorescence.

#### 3.2.7. Conclusions: Animal Studies

The animal studies evaluated the effects of vitamin D on muscle function and recovery. Overall, the studies demonstrated the importance of vitamin D in improving muscle cellular function and recovery of size after injury or in illnesses that led to muscle mass loss. These studies were cited as they directly addressed the role of vitamin D in muscle regeneration and repair. However, the literature has not explored the effects genetically deleting VDR and CYP27B1 have on skeletal muscle repair after injury. This is an important gap to address, as it would unravel the molecular mechanisms involved in skeletal muscle repair and regeneration. Furthermore, few studies have reported effects on muscle function. Understanding how vitamin D affects muscle function is paramount because it is what determines the risk of injury and quality of life of patients with vitamin D deficiency. Therefore, this review will also discuss studies that explored the role of vitamin D in muscle regeneration and recovery in humans.

### 3.3. Human Studies (Table 4)

#### 3.3.1. Introduction to Human Trials

Human studies were assessed to understand the effects of vitamin D on their skeletal muscle function and recovery after injury and how vitamin D deficiency affects muscle function and quality in the elderly population. Human studies investigating the role of vitamin D in muscle injury and regeneration are summarized in Table 4.

#### 3.3.2. Exercise-Induced Muscle Damage

Owens et al. investigated the effects of vitamin D on skeletal muscle function recovery and understood how vitamin D mediates muscle repair and regeneration [45]. A randomized controlled trial was conducted with 20 male volunteers who had borderline or insufficient levels of 25D (45 ± 25 nmol/l). Participants were asked to produce a maximum voluntary contraction (MVC) followed by eccentric exercise, which has been well-described to induce muscle damage. The eccentric exercise caused a ~30% decrease in MVC. After this injury, they were treated with 4000 IU/day of an oral vitamin D_3_ supplement or placebo for 6 weeks. Vitamin D treatment improved MVC at 48 h and 7 days after the muscle injury.

Bass et al. performed an experiment on human subjects who performed resistance exercise training for 20 weeks. Biopsies were performed on the vastus lateralis muscle 3–7 days after the final exercise, and pre- and post-exercise plasma 25D levels were measured. No significant correlations to circulating 25D levels were observed. However, the authors found that quadriceps *VDR* expression correlated to increases in participant lean mass, despite no correlation to changes in strength [43].

Barker et al. investigated the role of 25D levels in predicting muscle weakness after strenuous exercise. Serum 25D levels at baseline were 28.0 ± 2.5 ng/mL [68]. They then underwent intense exercise, and peak isometric forces were measured at 0, 24 h, 48 h, 72 h, and 7 days post-exercise. All subjects had significantly reduced peak isometric forces after intense exercise at all timepoints except 7 days. Subjects with lower pre-exercise serum 25D levels tended to have greater muscle weakness after intense exercise. To further emphasize the importance of 25D on muscle, Barker et al. also demonstrated that vitamin D supplementation could improve muscle function and attenuate weakness following intense exercise with improved peak isometric forces [69].

#### 3.3.3. Post-Surgery Studies

Brennan-Speranza et al. conducted a study to investigate the effects of vitamin D on skeletal muscle repair in patients with osteoarthritis [33]. The paper does not describe whether subjects were taking vitamin D supplements, but given the excellent average 25D levels, which are high for the age group in Melbourne, it is likely that many people were taking supplements. They found no significant differences in circulating 25D levels between OA and asymptomatic patients, while there were significant increases in VDR, vitamin D binding protein (DBP), and albumin levels in OA patients. They demonstrated clear expression of the VDR protein in human muscle.

Ryu et al. evaluated the prevalence of vitamin D deficiency among patients who underwent arthroscopic repair for a full-thickness rotator cuff tear, the relationship of vitamin D level with severity of the rotator cuff tear, and surgical outcomes after repair, including muscle structural integrity and function [70]. Overall, low serum 25D levels in patients were not related to the severity of rotator cuff tears, the degree of fatty infiltration in rotator cuff muscles, or any postoperative outcomes.

Rhee et al. also investigated the correlations between serum 25D levels and tissue vitamin D levels as well as with perioperative variables in patients who underwent arthroscopic rotator cuff repair [32]. Patients were mostly vitamin D deficient (serum 25D < 10 ng/mL) or insufficient (serum 25D > 10–20 ng/mL) pre-operation. Lower preoperative vitamin D was associated with lower serum and tissue vitamin D at 6 months and 1 year post-operation. Lower vitamin D levels were associated with greater muscle weakness 1 year post-operation. However, no correlation was found between the VDR, tissue and serum vitamin D levels, fatty degeneration in muscle, or healing failure of the rotator cuff post-surgery.

Ekinci et al. examined the combined effects of calcium β-hydroxy-β-methylbutyrate (CaHMB), vitamin D, and protein supplementation on the rate of wound healing, immobilization period, and muscle strength in female patients who underwent orthopaedic surgery [71]. The treatment group subjects were given a combination of 3 g CaHMB, 1000 IU Vitamin D, and 36 g protein supplements, twice per day, for 30 days post-operation. They found significantly shorter wound healing times in the combined treatment group compared to the controls. Furthermore, increased mobilization was observed on days 15 and 30 post-operation with combined treatment. Significantly higher hand grip strength was found with combined treatment on days 30. Together, these findings suggest that vitamin D in combination with CaHMB and protein supplementation accelerated wound healing after surgery, decreased patients’ immobilization period, and enhanced muscle strength, with no effects on BMI.

Shim et al. [34] investigated the effects of vitamin D supplementation on the expression of muscle vitamin D receptor (VDR) and cross-sectional area (CSA) in vitamin D-deficient and non-deficient patients with distal radial fracture (n = 18, females > 50 years old). Biopsies from patients’ forearm muscles were collected twice for this study, once during fracture repair and subsequently during hardware removal. Patients received 1000 IU of vitamin D per day for 8 months post-operation. Vitamin D supplementation normalized serum 25D levels in vitamin D-deficient patients after hardware removal. Furthermore, increased median VDR expression and CSA were observed after vitamin D supplementation, but this was only significant in vitamin D-deficient patients. Therefore, this study suggests vitamin D supplementation may increase expression of VDR and CSA in muscles in patients with fractures, although the effects may be more pronounced in those who are vitamin D deficient.

Cancienne et al. [35] determined whether there was any association between perioperative serum 25D levels and the failure of arthroscopic rotator cuff repair (RCR) requiring revision surgery. Patients that underwent surgery were included in the study (n = 982) and were divided into groups based on their serum 25D status. The rate of revision rotator cuff surgery rose significantly in patients with serum 25D deficiency (<20 ng/mL; 5.88%) compared to those with sufficient 25D levels (>30–<150 ng/mL; 5.88%). Despite serum 25D-insufficient patients (20–30 ng/mL) having a revision rate of 4.97%, this was not statistically different compared with serum 25D-sufficient controls.

Gordon et al. [72] compared lower-limb muscle size and strength in haemodialysis patients (n = 79) being treated with 1,25D (calcitriol at 0.5–9 mcg doses) or a 1,25 analogue (paricalcitol at 0.5–42 mcg doses) with those not receiving any treatment. Total thigh muscle cross-sectional area (CSA) was greater in patients that received active vitamin D (calcitriol or paricalcitol), and muscle strength outcomes were enhanced in these subjects after adjusting for both their age and gender.

Kim et al. [36] examined muscle biopsies in people undergoing rotator cuff surgery and compared people with vitamin D deficiency to those who were replete. They found significantly increased expression of proinflammatory genes IL1β and IL6 in both deltoid and supraspinatus muscles in people with deficiency. They confirmed by Western blotting that IL1β and IL6 proteins were both significantly increased. The study did not report on the effects of vitamin D deficiency on functional outcomes.

#### 3.3.4. Other Relevant Studies

Conzade et al. [73] examined associations of baseline 25D levels with sarcopenia and changes in muscle parameters and the role of parathyroid hormone (PTH) in these changes. Cross-sectional analyses showed that individuals with low baseline 25D levels tended to have a higher prevalence of sarcopenia and that vitamin D deficiency was associated with low grip strength, slow gait speed, and more time during the Time Up and Go (TUG) test. Prospective analysis performed after 3 years of follow-up demonstrated individuals with low baseline 25D had a 0.94% greater annual decrease in muscle mass index and a 3.06% greater annual increase in time to complete their TUG test. PTH did not play any role in sarcopenia development or changes in muscle parameters. Notably, although vitamin D status did not significantly change grip strength or gait speed, vitamin D insufficiency was associated with a higher risk of developing a low muscle mass index.

Bang et al. [44] aimed to understand the paraspinal muscle changes in relation to serum 25D levels in elderly women (n = 91, 60–69 years old). Serum 25D levels were measured, and subjects were subsequently stratified according to their vitamin D status: control, vitamin D insufficient, and vitamin D deficient. The study found that the vitamin D-deficient group had the lowest lumbar muscularity among all subjects, while they had the highest percentage of fatty infiltration in the paraspinal muscle.

Dzik et al. [74] performed a study to evaluate markers of oxidative stress and vitamin D receptor in paraspinal muscles in patients with lower back pain and vitamin D deficiency (n = 14), normal vitamin D levels (n = 10), and vitamin D supplementation (n = 14; 3200 IU/day) for 5 weeks. Patients with vitamin D deficiency showed elevated levels of lipid and protein peroxidation in muscle, although this was accompanied by increased antioxidant enzyme activity. Thus, the authors suggest that this compensatory increase in both processes in vitamin D-deficient patients may be absent in those given vitamin D supplementation, due to the role of vitamin D in eliminating markers of free radical damage and reducing oxidative stress in paraspinal muscles. In a second paper describing the same cohort, Dzik et al. [75] also highlighted the impact of vitamin D deficiency on these patients. Vitamin D deficiency was associated with increased oxidative stress and impaired mitochondrial function, as evaluated by decreased citrate synthase activity and PGC1a protein. Interestingly, although collectively the muscle atrophy data did not show any differences between groups, when stratified by gender, women with vitamin D deficiency had significantly increased expression of atrogin-1, a protein associated with atrophy.

Scott et al. [76] performed a study to describe associations between 25D, muscle parameters, and physical activity in older adults (n = 686, 50–79 years old). Their study found that subjects with low 25D levels tended to have significantly weaker leg strength, diminished muscle quality, and lower physical activity. Similarly, another larger-scale study performed by Gerdhem et al. [77] (n = 986; 75-year-old Swedish females) correlated lowered 25D levels with decreased muscle strength and physical activity.

Gronborg et al. reported a randomized, placebo-controlled trial of 1200 IU of vitamin D (30 µg) daily in fortified foods or placebo-foods for 12 weeks in 70 Pakistani and 66 Danish women living in Denmark [37]. The study did not test muscle injury/regeneration but did test muscle function. Unfortunately, the randomization was not successful, as the groups were significantly different on most parameters at baseline. Baseline grip strength and knee extension were significantly better in the active-treatment group. The groups also had significantly different baseline BMIs, body fat, and lean mass. The unsuccessful randomization makes interpreting the results challenging, but they did not find any differences with vitamin D supplementation.

Ceglia et al. performed a randomized, double-blind, placebo-controlled study of 4000 IU/day Vitamin D_3_ supplements to assess whether they altered muscle fiber cross-sectional area and intramyonuclear VDR concentrations in 21 mobility-limited women above 65 years old with serum 25D levels of between 22.5 and 60 nmol/L over 4 months [78]. The study found vitamin D treatment increased serum 25D levels, intramyonuclear VDR, and muscle fiber cross-sectional area at 4 months. While the authors investigated the effects of vitamin D supplementation on muscle morphology, they did not assess its effects on muscle function or changes in mobility. Furthermore, the limited sample size made it challenging for the authors to interpret their results, and they acknowledge that a larger sample size is needed to address the limitations of their study.

#### 3.3.5. Conclusions: Human Studies

Overall, the human studies show vitamin D has positive effects on muscle function and morphology. After muscle damage induced by strenuous exercise, vitamin D supplementation enhanced muscle strength with improvements in maximal voluntary contraction. On the other hand, it was surprising to find no correlations between low vitamin D levels, the severity of tears, and the rate of healing post-surgery, given that vitamin D plays an important role in improving muscle function and structure. However, improvements in recovery rates, muscle strength, and mobilization were only observed when vitamin D was given in combination with CaHMB and protein supplements. Vitamin D deficiency in elderly subjects was often associated with decreased muscle strength and muscle quality and an increase in the prevalence of sarcopenia.

**Table 4 nutrients-15-04377-t004:** Human studies of vitamin D effects on muscle function and structure.

Reference	Number of Subjects	Study Cohort Characteristics	Treatments/Surgeries Performed	Notes and Effects
[45], Owens	n = 20	18–30 years old, healthy male volunteers	4000 IU/day oral 1,25D supplementation following eccentric exercise + MVC	↑ MVC at 48 h and 7-days after muscle damage with 1,25D supplementation
[33], Brenna-Speranza	n = 19	>50 years old, males and females with knee osteoarthritis	Knee replacement surgery	No significant differences in 25D levels between OA and asymptomatic patients, ↑ VDR, ↑ DBP, ↑ albumin levels in OA patients
[43], Bass	n = 37	18–35 years old, healthy males and females	20 weeks resistance exercise training	↑ muscle hypertrophy following long-term resistance training exercise
[73], Conzade	n = 975	63–93 years old	No treatments or surgeries performed	↓ grip strength, ↓ gait speed, ↑ time during TUG test, ↑ sarcopenia in subjects with low serum 25D levels
[70], Ryu	n = 91	50–60 years old	Arthroscopic rotator cuff repair	Low serum 25D levels in patients were not correlated to the severity of rotator cuff tears
[32], Rhee	n = 36	40–80 years old	Arthroscopic rotator cuff repair	No correlation between tissue and serum vitamin D levels with fatty degeneration in muscle and healing failure post-surgery.
[44], Bang	n = 91	60–69 years old, females	No treatments or surgeries performed	↓ lumbar muscularity, ↑ percentage of fatty infiltration in Vitamin D deficient subjects
[74], Dzik (2018)	n = 38	average 48.2 years old, males and females with lower back pain	Vitamin D supplementation (3200 IU/day) for 5 weeks	↓ markers of free radical damage of lipids and proteins, ↓ oxidative stress in paraspinal muscle with vitamin D supplementation
[75], Dzik (2019)	n = 14	average 45.2–51.2-year-old patients with lower back pain	Vitamin D supplementation (3200 IU/day) for 5 weeks	↑ muscle atrophy, ↑ oxidative stress, ↓ mitochondrial function with vitamin D deficiency
[76], Scott	n = 686	50–59 years old	No treatments or surgeries performed	↓ leg strength, ↓ muscle quality, ↓ physical activity in patients with low 25D levels
[77], Gerdhem	n = 986	75 years old, females	No treatments or surgeries	↓ muscle strength and physical activity
[72], Gordon	n = 79	49 years old	1,25D treatment with calcitriol (0.5 mcg-9 mcg) or paricalcitol (0.5–42 mcg)	↑ thigh muscle cross-sectional area, ↑ muscle strengths with active vitamin D (calcitriol or paricalcitol) treatment
[68], Barker (2013a)	n = 14	32 years old, males and females	Intense-stretch shortening contraction	↓ peak isometric forces after intense exercise at all time-points (except day 7) ↓ muscle strength in subjects with low pre-exercise 25D levels
[69], Barker (2013b)	n = 28	30 years old	4000 IU Cholecalciferol (Vitamin D) for 35 days	↑ muscle function and strength following intense exercise with Vitamin D supplementation
[71], Ekinci	n = 75	≥65 years old, females	3 g CaHMB + 1000 IU Vitamin D + 36 g Protein supplementation following orthopaedic surgery	↓ wound healing time, ↑ mobilisation, ↑muscle strength with combined treatment post orthopaedic surgery
[34], Shim	n = 18	>50 years old, females	Distal Radial Fracture repair + 1000 IU Vitamin D supplementation	↑ medical VDR and muscle CSA after vitamin D supplantation Normalized serum 25D levels in vitamin D deficient patients post-surgery
[35], Cancienne	n = 982	>50 years old	Arthroscopic rotator cuff repair	↑ rate of revision rotator cuff surgery in patients with vitamin D deficiency
[36], Kim	n = 24	41–75 years old	Arthroscopic rotator cuff repair	↑ IL1β and IL6 in deltoid and supraspinatus muscles in vitamin D deficient subjects. No functional outcomes were assessed.
[37], Gronborg	n= 136	18–50 years old	Vitamin D fortified diet	↑ grip strength and knee extension in active treatment groups
[78], Ceglia	n = 21	≥65 years old, females	4000 IU/day oral Vitamin D3 supplementation	↑ serum 25D levels, ↑ intramyonuclear VDR, ↑ muscle fiber CSA at 4 months post-supplementation

## 4. Interaction between Vitamin D and Physical Exercise

Due to the association between vitamin D deficiency and muscle weakness, we examined the relationship between vitamin D and exercise.

Most studies examined either professional athletes or older adults. Some studies found that vitamin D supplementation improved physical activity levels [79,80], while others found no associations [81,82,83]. Interestingly, one study has also suggested bouts of physical exercise increase serum 25D levels [84]. Overall, the exact relationship between vitamin D levels and exercise remains unclear, and whether vitamin D supplementation should be given to individuals to improve their exercise levels is widely debated. Inclusion of vitamin D testing in studies aiming to improve people’s exercise patterns would be of interest, with before and after levels.

As some intervention studies include a large number of participants, this may answer this question. The benefits of having enough participants in these studies are that they represent a large portion of the population and mitigate the risk of false-negative or false-positive findings. Therefore, given the importance of optimal muscle function in exercise, the relationship between vitamin D and physical exercise levels should be further investigated in the future.

## 5. Discussion

Muscle regeneration is a homeostatic process that allows skeletal muscles to restore structural and functional integrity after injury [16]. The process is mediated by skeletal muscle stem cells, known as satellite cells [16,17,85]. These cells are located between the basal lamina and sarcolemma of myofibers and only become mitotically activated during muscle growth or damage [16,17]. Vitamin D is a steroid hormone responsible for calcium and phosphate homeostasis, which are integral to the maintenance of skeletal muscle function and structure. Therefore, there has been growing interest in exploring the role of vitamin D in skeletal muscle regeneration.

The in vitro studies provided insights into the molecular mechanisms of vitamin D in skeletal muscle cells at different stages of myogenesis. The studies performed confirmed the presence of vitamin D receptor (VDR) and vitamin D signaling components, CYP24A1 and CYP27B1, in skeletal muscle cells during the different stages of myogenesis [8,42,50]. They also found that 1,25D promoted protein anabolism and hypertrophy increased the cross-sectional area of skeletal muscle cells and explored its molecular mechanisms in skeletal muscles [8,38,55]. 1,25D was also found to enhance the effects of leucine and insulin, which promote protein anabolism in muscle cells [55,86,87]. Overall, the studies suggest that it played a regulatory role in muscles due to its different effects in various stages of myogenesis. In satellite cells, 1,25D promoted myogenesis with decreased expression of myogenic regulatory factors and myostatin found at various 1,25D doses [13,38,51]. However, an increase in myogenic regulatory factor expression and myogenin protein was found with 1,25D treatment in proliferating myoblasts [27,52]. They were also found to play a role in regulating angiogenesis [39], cell cycle regulators during different stages of skeletal muscle development [8,61], the ubiquitin-proteasome pathway [64], the Akt signaling pathway [55,56,57], and mitochondrial function [58,62,88]. Although these studies provide insight into the mechanism of vitamin D in muscle cells derived from mice and humans, they cannot address its direct role in muscle regeneration. Hence, it is critical to also assess animal studies that examine the relationship between vitamin D and skeletal muscle regeneration.

The animal studies evaluated the effects of vitamin D on muscle function and recovery. The studies showed VDR and CYP27B1 played an important role in maintaining muscle function and structure, with genetic deletion of those genes having a negative impact on these parameters [12,24,28,66]. The importance of vitamin D in skeletal muscle recovery upon induced muscle injury has also been highlighted, with many studies noting an increase in proliferation and reduction in apoptosis, accompanied by elevated *Vdr* and *Cypb1* expression [31,42]. In addition, many studies also demonstrate the benefits of vitamin D diets and supplementation on muscle health in healthy and disease-related animal models, which further emphasize its potential therapeutic benefits [15,25,26,27,29,31,67]. Finally, animal studies have also unveiled potential mechanisms for vitamin D in promoting muscle regeneration, with suggested roles in the ubiquitin-proteasome pathway, regulation of mitochondrial biogenesis and function, and angiogenesis [50,67]. However, the vitamin D dosage administered in animal models in two studies was extremely high for humans [26,31]. Nevertheless, the studies helped address the importance of vitamin D in improving muscle function and recovery after injury or in illnesses that led to muscle mass loss.

In view of the positive effects of vitamin D on skeletal muscle function and structure in animal models, this review explored studies that investigated the role of vitamin D in muscle regeneration and recovery in humans. Vitamin D deficiency was shown to have negative effects on muscle function and quality, especially in the elderly population. Hence, it was surprising to find that most studies reported no associations between low serum vitamin D levels and the severity of rotator cuff tears and healing failure, given its importance in muscle function and structure. However, it seemed to help reduce healing time, improve mobilization, and improve muscle strength when given in combination with CaHMB and protein supplements in patients who have undergone surgery. Overall, it is possible that vitamin D signaling may not play as important a role in muscle tears (such as those seen in rotator cuff injuries) as it plays in muscle injuries where the overall shape of the muscle is relatively preserved (exercise, toxin, or crush-induced injuries).

We note that many of the positive studies involved supplementation with very high doses of vitamin D. In many studies, subjects were treated with 1000–4000 IU of vitamin D/day. It is possible that any muscle benefits of vitamin D may require greater vitamin D availability than effects on calcium and phosphate homeostasis.

### Comparisons between Animal and Human Studies

Both the animal and human studies demonstrate vitamin D plays an important role in skeletal muscle function and structural integrity, which suggests that vitamin D is important for skeletal muscle repair and regeneration.

The key benefit of utilizing animal models is the ability to assess the effects of vitamin D deficiency, crush injury, and genetically deleting vitamin D signaling intermediates on muscle morphology and function. This includes being able to interrogate the changes in the expression of genes related to the cell cycle, calcium handling, and myogenesis. Overall, these studies have demonstrated the impairments induced by vitamin D deficiency or VDR deletion and, conversely, the benefits of high levels of vitamin D supplementation. At a mechanistic level, vitamin D deficiency has been shown to decrease calcium handling, cell cycle, and myogenic regulatory factors and result in decreased protein anabolism and decreased muscle fiber size and weight. In addition, a high dosage of vitamin D after a crush injury was able to reduce apoptosis in rat muscles.

On the other hand, given that it is unethical to make humans vitamin D deficient, human studies typically assess the positive effects of vitamin D on skeletal muscle function and healing time post-surgery. There is strong evidence showing vitamin D supplementation improves muscle strength after exercise-induced muscle damage, with increased muscle fiber cross-sectional area, improvements in muscle mitochondrial function, and decreased muscle wasting observed. Although the mechanistic pathways implicated in animal models have not quite been demonstrated in human studies, the similarities observed in enhanced muscle repair following injury suggest that the aforementioned pathways may be underpinning the improvements observed in humans.

## 6. Strengths and Limitations

The literature has established the effects of vitamin D treatment on myocyte proliferation and differentiation and on multiple processes involved in maintaining muscle integrity, such as angiogenesis and mitochondrial respiration [39,88]. These findings are beneficial to the field as they provide us with a deeper understanding of the role vitamin D has in muscle maintenance and potentially in its repair and regeneration. Moreover, it has explored the effects of genetically deleting VDR and CYP27B1 on muscle function, morphology, and gene expression in mouse models. It has also explored the effects of vitamin D repletion on animals with muscle injuries, which directly addresses our review question. In addition, there are extensive studies on the effects of vitamin D deficiency on muscle strength and quality in elderly individuals, which is the population most severely affected by vitamin D deficiency. Finally, there is compelling evidence for the therapeutic potential of vitamin D-mediated muscle recovery after strenuous exercise and surgery in humans, which directly addresses the role of vitamin D in skeletal muscle repair and regeneration.

However, we believe that more in vitro studies exploring the effects of vitamin D on satellite cells could be performed, as these cells play an important role in muscle repair and regeneration by initiating myocyte proliferation and differentiation [19,89]. There is also a need to explore the direct effects of genetically deleting VDR and CYP27B1 on muscle repair and regeneration, which can be achieved by inducing muscle damage (crush injury, toxin-induced) in mouse models with this genetic modification. Also, given the severity of vitamin D deficiency-induced muscle function in the elderly population, there also needs to be more focus on performing the animal studies reported in the literature in aged mice. Finally, we also acknowledge that other nutrients (e.g., omega-3) play a role in improving muscle strength and structure, which has not been explored in this review [90,91]. Hence, it can be another topic for deeper exploration in the future.

## 7. Conclusions

In summary, the studies we reviewed confirm the importance of vitamin D in skeletal muscle function and structural integrity and have explored potential molecular mechanisms for vitamin D in aiding skeletal muscle regeneration and repair. This has been assessed in cell, animal, and human studies. Overall, vitamin D inhibited myocyte proliferation, enhanced myocyte differentiation, mitochondrial respiration, angiogenesis, muscle function, and size, and it helped reduce post-surgery healing times and the rate of revision rotator cuff surgeries. While the mechanistic pathways established in cell and animal studies have not yet been shown in humans, they provide a strong foundational basis for avoiding 25D deficiency in the context of muscle injury. In combination with human studies that illustrate the importance of vitamin D in restoring skeletal muscle function and structure, particularly after injury and illnesses that induce muscle damage, we highlight the importance of utilizing 25D as a therapeutic in patients with sarcopenia and muscle injury. Therefore, we recommend consideration of vitamin D supplementation with at least 1000 IU daily in people with muscle injury.

## Figures and Tables

**Figure 1 nutrients-15-04377-f001:**
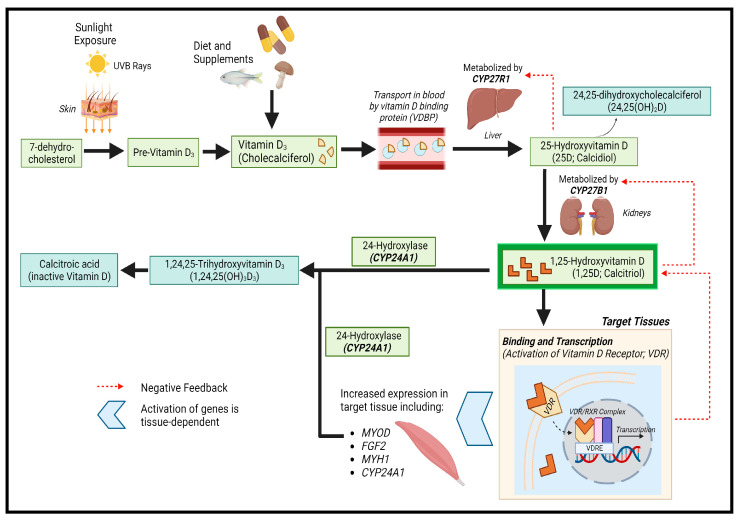
Vitamin D synthesis pathway. Created with BioRender.com.

**Figure 2 nutrients-15-04377-f002:**
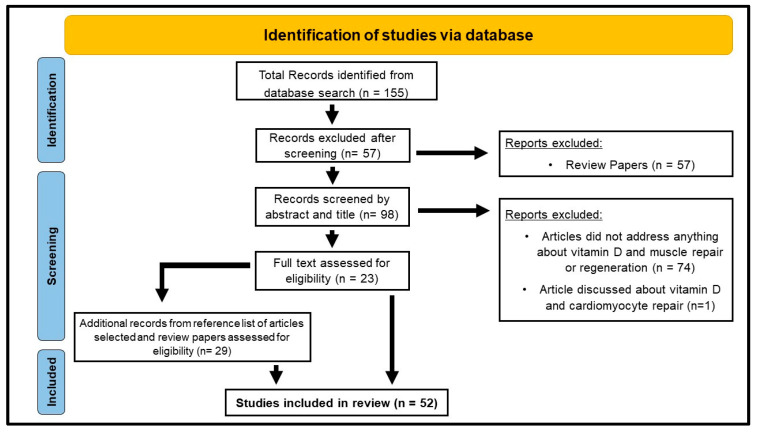
Study selection flow diagram adapted from the Preferred Reporting Items for Systematic Reviews and Meta-Analyses (PRISMA) guidelines [22].

**Table 1 nutrients-15-04377-t001:** Literature search inclusion and exclusion criteria.

Inclusion Criteria	Exclusion Criteria
Original research articles (in vitro and/or in vivo studies using animal models and human participants) that discussed the role of vitamin D in muscle regeneration; Clinical trials and observational studies that discussed the role of vitamin D in muscle regeneration; Articles had to be written in English; Articles that discussed the role of vitamin D in combination with other compounds in muscle regeneration.	Review articles (these were checked for potential additional references); Articles that focus on the role of other compounds (e.g., omega-3, proteins) on muscle function and structural integrity; Articles that discuss optimal vitamin D therapeutic doses without any context of muscle regeneration; Articles that discussed the role of vitamin D in other tissues (e.g., bone) but not muscle.

## Data Availability

Not applicable.

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
