# Peer review of "The Role of Vitamin D in Skeletal Muscle Repair and Regeneration in Animal Models and Humans: A Systematic Review"

_nutrients, 2023, doi:10.3390/nu15204377_

Round 1

Reviewer 1 Report

The manuscript was prepared very well. The introduction section justifies the purpose of the study. I congratulate the authors for the preparation of the manuscript

I would like to congratulate the authors for the structure of the manuscript and all the research carried out. It is highly publishable. However, there are some concerns, in part important, so the review articles need revision, see below.

Title

include the reviewed studies in the title since 2 categories of human and animal studies are established.

Abstract

·       Revise the abstract and include more concrete results.

Introduction

·       Why is this study considered relevant?

·       Why is this study necessary?

·       There are many recent reviews in this field, which contributes yours, please clarify

Methods and Results

·       If it is a systematic review it should contain the PRISMA rules

·       I recommend passing the search string to Appendix

·       could you add the PICOS question?

·       I recommend your registration in PROSPERO

-        Discussion

·       Include a section on strengths / limitations.

·       Describe in a section if there are differences between the results of studies in humans and animals

·       What mechanisms of action support these findings?

·       What does this article contribute to, the authors should make their own assessment and include their own discussion of the results shown in the manuscript.

Conclusion

In the Conclusion section, state the most important outcome of your work. Do not simply summarize the points already made in the body — instead, interpret your findings at a higher level of abstraction. Show whether, or to what extent, you have succeeded in addressing the need stated in the Introduction (or objectives).

must be reviewed by a native person

Reviewer 2 Report

In the article under review, Agoncillo et al. explore the role of Vitamin D in the regeneration and repair of muscle tissue. They scrutinize a wide array of studies, including in-vitro experiments, animal models, and human studies, emphasizing the crucial role of Vitamin D in muscle function and structure. Despite the comprehensive analysis, the review could benefit from a more critical evaluation of the studies, a discussion on potential mechanisms of action, and consideration of other factors influencing muscle repair and regeneration.

The following concerns need to be addressed:

  1. The authors should broaden the scope of the review by incorporating the potential role of other nutrients known to contribute to muscle health, such as protein, omega-3 fatty acids, and certain vitamins and minerals, and their potential interaction with Vitamin D.
  2. The review should also delve into the interaction between Vitamin D status and physical activity, given the significant role of physical activity in muscle health and its potential impact on the effects of Vitamin D on muscle repair and regeneration.
  3. Based on the review's findings, the authors could offer practical recommendations for clinicians, especially for individuals with muscle injuries or health conditions affecting muscle tissue.
  4. Lastly, the authors could propose areas for future research based on the identified gaps in the current literature. This could include specific study designs, target populations, or outcomes that should be explored in future studies.

Reviewer 3 Report

Abstract:

Line 11-12: Consider rephrasing for clarity: "Vitamin D deficiency, prevalent worldwide, is linked to muscle weakness, sarcopenia, and falls."

Line 13-14: The phrase "relationship between vitamin D and skeletal muscle regeneration" can be made more concise: "relationship between vitamin D and muscle regeneration."

Line 17-19: The sentence structure is a bit convoluted. Consider: "Animal studies, primarily in mice, demonstrate vitamin D's positive effects on skeletal muscle function, such as improved grip strength and endurance. These studies encompass vitamin D diet research, genetically modified models, and disease-related mouse models."

Line 26-28: The conclusion can be more concise: "In summary, vitamin D plays a crucial role in skeletal muscle function, structure, and regeneration, potentially offering therapeutic benefits for musculoskeletal diseases and post-operative recovery."

Introduction:

Line 43-44: Repetition from the abstract. Consider rephrasing to avoid redundancy.

Line 50-59: This section provides a detailed biochemical pathway of vitamin D synthesis and action. Ensure that all the references are correctly cited and consider using a figure or diagram to illustrate this pathway for better clarity.

Line 73-81: The distinction between whole-body VDRKO mice and mVDR mice is crucial. Consider emphasizing this difference more clearly, perhaps with subheadings or highlighted text.

Material and Methods

Search Strategy:

Line 93-94: The search strategy is a good start, but it might be too narrow. Consider expanding the search terms to capture more relevant articles. For instance, include synonyms or related terms for "muscle" like "myocyte" or "skeletal muscle."

Specify which fields were searched (e.g., title, abstract, keywords) to provide clarity on the search strategy.

Databases:

Only PubMed is mentioned. Consider searching additional databases like Scopus, Web of Science, or EMBASE to ensure a comprehensive literature search.

Inclusion and Exclusion Criteria:

Line 95-96: The phrase "the following inclusion and exclusion criteria was applied" should be corrected to "the following inclusion and exclusion criteria were applied."

Under Inclusion Criteria:

Specify the types of articles included (e.g., original research articles, clinical trials, observational studies).

Consider specifying the study populations (e.g., human, animal models) and study designs (e.g., randomized controlled trials, cohort studies) that were included.

Under Exclusion Criteria:

The criteria "Articles that do not address anything about vitamin D and muscle repair or regeneration" seems redundant given the search terms and other criteria. Consider removing or rephrasing for clarity.

The criteria "Articles that only discuss optimal vitamin D therapeutic doses" could be made clearer. Does this mean articles that solely discuss dosing without any context of muscle regeneration?

PRISMA Flow Diagram:

It's noted that you've included a flow diagram, but it doesn't adhere to the PRISMA format. The PRISMA flow diagram should be structured to visually represent the study selection process, showing the number of articles identified, screened, eligible, and included in the review. Furthermore, the flow diagram should be placed in the "Results" section, not in the "Materials and Methods."

Data Extraction:

There's no mention of how data were extracted from the selected studies. Specify who extracted the data, whether it was done in duplicate, and what information was extracted (e.g., study design, sample size, main findings).

Quality Assessment:

There's no mention of how the quality or risk of bias of the included studies was assessed. Consider using a standardized tool or checklist appropriate for the types of studies included in the review.

PICO Framework:

The PICO (Population, Intervention, Comparison, Outcome) framework is not explicitly mentioned. While this is a systematic review and not a clinical question, it would be helpful to structure the review around:

Population: Which populations were studied (e.g., humans, specific animal models)?

Intervention: What was the role or effect of vitamin D?

Comparison: Was there a comparison group? If so, what was it?

Outcome: What outcomes related to muscle regeneration were measured?

Registration Number:

It's essential to provide the registration number of the systematic review. This number is crucial for transparency and to ensure that there's no duplication of systematic reviews on the same topic. Please include this in the "Materials and Methods" section.

Results:

The text provides a comprehensive review of studies conducted on the role of vitamin D in muscle function and regeneration, spanning from in vitro experiments, animal models, to human clinical trials. While the content is rich and informative, the current structure could benefit from some reorganization to enhance clarity and reader comprehension.

Structure:

The current structure divides the studies into two main categories: animal studies and human studies. Within these categories, there are subsections detailing different types of studies or contexts (e.g., exercise-induced muscle damage, post-surgery studies, etc.). However, the in vitro studies seem to be embedded within these sections, which might be causing some confusion.

Suggested Structure:

Introduction: Brief overview of the role of vitamin D in muscle function and the significance of studying it.

In Vitro Studies:

- Introduction to in vitro experiments.

- Main findings and their implications.

Animal Studies:

- Introduction to animal models.

- Muscle injury.

- Vitamin D deficiency.

- VDR overexpression.

- Conclusions from animal studies.

Human Studies:

- Introduction to human trials.

- Exercise-induced muscle damage.

- Post-surgery studies.

- Other relevant studies.

- Conclusions from human studies.

General Discussion: Compare and contrast findings from in vitro, animal, and human studies, discuss clinical implications, and provide recommendations for future research.

Conclusion: Summarize the main findings and their significance in the field of medicine and biology.

Discussion in Each Section:

Currently, each subsection ends with a sort of conclusion that summarizes the findings. However, a deeper discussion within each subsection would be beneficial. This discussion should not only summarize results but also compare them with existing literature, discuss their implications, and highlight any limitations.

Conclusion:

It is indeed lengthy and might be overwhelming for readers. The authors have delved into the specifics of each study type (in-vitro, animal, and human) in the conclusion, which can be streamlined for clarity and brevity.

The conclusion should be concise, summarizing the main findings without delving too deeply into specifics. Specific details are better suited for the main body of the article.

- Focus on the primary takeaways from each section (in-vitro, animal, and human studies) without getting into the details of each study.

- Emphasize the clinical or practical implications of the findings, especially regarding vitamin D supplementation.

- End with clear recommendations based on the findings. This will provide readers with actionable insights.

Round 2

Reviewer 1 Report

I suggest some higlihths of the manual given the number of results shown.

In addition, the number of abbreviations makes reading difficult, I suggest a glossary of abbreviations

In addition to the results obtained, what would be its potential application? What perspectives does this study have for real field practice?

requires minor changes and suggests review by a native

Reviewer 2 Report

The authors have addressed all the concerns satisfactorily.

Reviewer 3 Report

The authors have significantly improved the quality of their article. However, there are still some aspects that need to be addressed.

Firstly, for a systematic review, it is advisable to search in at least three databases. The authors have searched in two. I would recommend adding one more, such as SCOPUS or Cochrane.

Additionally, the flow diagram is not clear. Did the authors introduce 27 articles from the reference lists of other articles? Were these articles included in the studies? Why do the tables account for a total of 55 articles when, in theory, you have a total of 48? This needs clarification.

Articles with the same nomenclature should be differentiated in some way. For instance, García (38) and García (51).

The authors claim to have included a bias risk section. However, I was unable to locate it in the paper. Depending on the included articles, this should be conducted using tools like ROBINS I, ROBINS E, ROB 2.0, etc.

The authors state that the systematic review was registered in PROSPERO retrospectively. This should never be done, as it does not verify if the review was conducted beforehand. The authors must provide the registration number before proceeding with the review of this paper.
